# Identification and Expression Analysis of the WOX Transcription Factor Family in Foxtail Millet (*Setaria italica* L.)

**DOI:** 10.3390/genes15040476

**Published:** 2024-04-10

**Authors:** Lizhang Nan, Yajun Li, Cui Ma, Xiaowei Meng, Yuanhuai Han, Hongying Li, Mingjing Huang, Yingying Qin, Xuemei Ren

**Affiliations:** 1College of Agriculture, Shanxi Agricultural University, Taigu, Jinzhong 030800, China; nlizhang328@163.com (L.N.); lyajun0704@163.com (Y.L.); aui0317@163.com (C.M.); mxw_0901@163.com (X.M.); swgctd@163.com (Y.H.); hongying1964@163.com (H.L.); 13653650538@163.com (M.H.); 2College of Life Sciences, Shanxi Agricultural University, Taigu, Jinzhong 030800, China

**Keywords:** foxtail millet, WOX transcription factor family, expression pattern, plant hormones

## Abstract

WUSCHEL-related homeobox (WOX) transcription factors are unique to plants and play pivotal roles in plant development and stress responses. In this investigation, we acquired protein sequences of foxtail millet *WOX* gene family members through homologous sequence alignment and a hidden Markov model (HMM) search. Utilizing conserved domain prediction, we identified 13 foxtail millet *WOX* genes, which were classified into ancient, intermediate, and modern clades. Multiple sequence alignment results revealed that all WOX proteins possess a homeodomain (HD). The *SiWOX* genes, clustered together in the phylogenetic tree, exhibited analogous protein spatial structures, gene structures, and conserved motifs. The foxtail millet *WOX* genes are distributed across 7 chromosomes, featuring 3 pairs of tandem repeats: *SiWOX1* and *SiWOX13*, *SiWOX4* and *SiWOX5*, and *SiWOX11* and *SiWOX12*. Collinearity analysis demonstrated that *WOX* genes in foxtail millet exhibit the highest collinearity with green foxtail, followed by maize. The *SiWOX* genes primarily harbor two categories of *cis*-acting regulatory elements: Stress response and plant hormone response. Notably, prominent hormones triggering responses include methyl jasmonate, abscisic acid, gibberellin, auxin, and salicylic acid. Analysis of *SiWOX* expression patterns and hormone responses unveiled potential functional diversity among different *SiWOX* genes in foxtail millet. These findings lay a solid foundation for further elucidating the functions and evolution of *SiWOX* genes.

## 1. Introduction

Transcription factors encompass a wide range of classifications and functions. They possess the capacity to bind to the promoters of target genes, thereby either activating or repressing their transcriptional activity. These factors orchestrate the specific expression of target genes across diverse tissues, cells, and under various external conditions, governing vital life processes through cascades of transcriptional regulation. The WUSCHEL-related homeobox (WOX) transcription factor family, unique to plants, constitutes a distinct subfamily within the eukaryotic transcription factor homeobox family. Its members exhibit diverse biological functions, primarily contributing to the dynamic equilibrium of plant stem cell division and differentiation, embryonic and post-embryonic development, hormone signal transduction, and responses to crop stress [1,2,3]. Proteins within the WOX family feature a conserved homeodomain (HD) structure composed of 60 amino acid residues, with WUS additionally harboring a tyrosine (Y) residue. These amino acids are arranged into a “helix-loop-helix-turn-helix-loop-helix” structure, where the “helix-loop-helix-turn-helix” formed by the second and third helices can bind to specific DNA sequences, thereby regulating the transcription of downstream genes [4]. Research on the classification of WOX family members primarily focuses on their evolutionary origins. By reconstructing phylogenetic trees of plant WOX using homologous HD domain protein sequences, the phylogenetic tree of plant WOX can be categorized into three clades: Modern, intermediate, and ancient. Bioinformatics analysis of 350 *WOX* genes in 50 plants revealed that WOX members in lower plants originate solely from the ancient clade, while WOX members in higher plants are exclusively found in the modern clade, with the intermediate clade emerging in vascular plants. This study suggests that the intermediate clade and the evolutionary clade may have evolved from the duplication and differentiation of ancient clade members [5,6,7]. The conserved HD structure of the WOX transcription factor family remains highly conserved across different species. In addition to the HD domain, it also contains a WUS-box domain, exclusive to modern clade members, positioned at the carboxy-terminal end of the homologous domain. The amino acid sequence is in the form of T-L-X-L-F-P-X-X, with X representing any amino acid. Non-modern clade WOX family members exhibit variations at this position. The WUS-box domain plays a critical role in regulating the *WUS* gene’s control of stem cell characteristics in shoot apical meristems and floral meristem morphology [3,8,9]. Some family members feature carboxy-terminal ethylene-responsive element binding factors (ERF)-related domains, known to be involved in transcriptional repression [10].

In the model plant *Arabidopsis thaliana*, a total of 16 WOX family members have been identified, comprising *AtWUS* and *AtWOX1* to *AtWOX15*. Previously documented WOX transcription factor family members are closely associated with regulating stem cell fate, as well as initiating and developing tissues and organs [7]. Among the 16 WOX members, the modern clade encompasses *WUS* and *WOX1* to *WOX7*; the intermediate clade comprises *WOX8*, *WOX9*, *WOX11*, *WOX12*, and *WOX15*; while the ancient clade includes WOX10, WOX13, and WOX14 [7,8]. *WUS* was the pioneering gene discovered in the WOX family, acting as a pivotal regulator of stem cell regulation in shoot apical meristems. Through EMS mutagenesis, the *wus* mutant exhibited defective shoot apical meristems, reduced floral organ numbers, and missing pistils [11]. Subsequent investigations unveiled that in mutants with *WUS* loss of function, *WUS* plays a crucial role in maintaining stem cell characteristics and quantity, while inhibiting stem cell differentiation [12], Moreover, *WUS* is implicated in ovule and anther development in Arabidopsis [13,14], with similar roles observed in rice and maize [15]. In Arabidopsis plants overexpressing *TaWUS*, mutants exhibit early flowering and increased numbers of floral organs, including petals, sepals, and flowers, while the number of stamens and pistils remains unchanged [16]. Genetic and molecular studies have revealed that in the regulatory network of shoot apical meristems in Arabidopsis, *WUSCHEL* (*WUS*) interacts with *CLAVATA* (*CLV*). Assisted by related transport proteins, *WUS* relocates from the tissue center to the stem cell area via intercellular connections, stimulating the expression of the *CLV3* gene to uphold stem cell characteristics. Once *CLV3* binds to its receptor protein CLV1, it dampens *WUS* transcription levels, facilitating stem cell differentiation and thus forming a *CLV*-*WUS* feedback loop to regulate the dynamic balance of stem cells in Arabidopsis shoot apical meristems [17,18,19,20]. This feedback loop can also interact with plant hormones such as cytokinins and auxins to collectively regulate meristem development [21,22]. Furthermore, *AtWOX5* exhibits specific expression in the root apical meristem, where it suppresses the differentiation of root apical stem cells, ensuring a constant stem cell count and fostering the growth of root apical meristems [23], *CLE40*, acting through receptors ACR4 and CLV1, inhibits the expression of *AtWOX5*, thereby promoting stem cell differentiation, suggesting the presence of a regulatory pathway akin to the shoot apical meristem (SAM) in Arabidopsis’ root apical meristem. During seed development in Arabidopsis [23,24]. *AtWOX2* and *AtWOX8* are co-expressed, with *AtWOX2* playing a crucial role in initiating shoot apical stem cells in embryos [25,26]. *AtWOX3* participates in regulating the lateral axis development of flowers, with mutants displaying loss or narrowing of petals [27], *AtWOX11* and *AtWOX12* collectively influence root meristem development [28], Similarly, *AtWOX13* and *AtWOX14* play roles in regulating both flower and root development, with *AtWOX13* primarily expressed in the stigma and pistil [29]. Recent research has unveiled *AtWOX13* as a pivotal regulatory factor in Arabidopsis callus tissue formation. Following explant injury, AtWOX13 is rapidly induced, partially reliant on the activity of the AP2/ERF transcription factor WOUND-INDUCED DEDIFFERENTIATION 1 (WIND1). Subsequently, *AtWOX13* directly upregulates WIND2 and WIND3, thereby promoting cell proliferation and organ regeneration [30].

In the growth and development of crops such as rice, maize, and tomato, WOX transcription factors also play pivotal roles. For instance, *OsWOX1/MOC3* serves as a homolog of the Arabidopsis *WUS* gene in rice, whereas *MOC1* encodes a plant-specific GAI-RGA-SCR (GRAS) family protein. *FON1*, on the other hand, acts as a homolog of Arabidopsis *CLV1* in rice. *MOC1* activates *FON1* expression by binding to its promoter, and further investigations have demonstrated that MOC1 can interact with MOC3, acting as a co-activator to enhance *FON1* expression [31,32,33,34]. In maize, *ZmWUS1* and *ZmWUS2* function as homologs of Arabidopsis *WUS*. It has been observed that the maize mutant *Bif3* contains an enhancer region with multiple binding sites for type B response regulators (B-type RR), which significantly upregulates the expression of *ZmWUS1-B*, resulting in excessive development of meristematic stem cells in the inflorescence [35]. In rice, *OsNS*, and in maize, *ZmNS1*/*ZmNS2*, serve as homologs of *AtWOX3*. *OsNS* exhibits predominant expression in rice leaf primordia, young leaves, and flowers, while *ZmNS1*/*ZmNS2* ensure the proper phenotype of maize leaves and floral organs [36,37,38,39]. Overexpression of the tomato gene *SlWOX13* significantly influences the expression of genes associated with leaf development, meristem development, and trichome development. Additionally, *SlWOX13* is implicated in tomato plant gibberellin synthesis and response, thereby affecting normal plant growth and development [40]. Recent studies have unveiled that *SlWOX13* directly activates the expression of multiple genes involved in ethylene synthesis and signaling, as well as carotenoid biosynthesis, thereby positively regulating tomato fruit ripening [41].

With the completion of foxtail millet whole-genome sequencing [42], numerous gene families of foxtail millet have been identified and subjected to bioinformatics analysis. As a model crop of C_4_ plants [43], foxtail millet has recently been found to possess *YUC* [44], *CLE* [45], and *GRF* [46] families associated with its growth and development. Despite extensive studies on the WOX gene family in various species such as Arabidopsis [8], rice [47], sorghum [47], cotton [48], wheat [16], maize [47,49], mung bean [50], sunflower [51], cucumber [52], melon [53], and *Medicago sativa* [54], there have been no reports on the *WOX* gene family in foxtail millet. This study identified 13 foxtail millet *WOX* genes and constructed a phylogenetic tree comprising foxtail millet (13), Arabidopsis (16), rice (13), maize (21), tomato (10), and green foxtail (13). Through RNA-seq data analysis, the study scrutinized the protein spatial structure, gene structure and conserved motifs, *cis*-acting elements, chromosome location, tissue-specific expression patterns, and response to different plant hormones of the *WOX* gene family in foxtail millet. These findings suggest that the *WOX* gene family plays a crucial role in the growth and development of foxtail millet and its response to various plant hormones, laying the groundwork for further elucidating the functions of *WOX* genes in foxtail millet.

## 2. Materials and Methods

### 2.1. Identification and Phylogenetic Analysis of WOXs in Foxtail Millet

The genomic, proteomic, coding sequence, and GFF annotation data of foxtail millet were downloaded from the foxtail millet multi-omics database MDSI (http://foxtail-millet.biocloud.net/home, accessed on 4 December 2023). Whole-genome data of Arabidopsis, tomato, rice, maize, and green foxtail were obtained from the Phytozome database (https://phytozome-next.jgi.doe.gov/, accessed on 4 December 2023). The HMMER model (PF00046) of the *WOX* gene family was downloaded from the Pfam database (http://pfam.xfam.org/, accessed on 4 December 2023). TBtools (v2.034) [55] software was used to perform HMM searches in the total protein sequences of foxtail millet, filtering for *SiWOX* genes. Protein sequences of *WOX* genes in Arabidopsis were downloaded from the TAIR database (https://www.arabidopsis.org/, accessed on 4 December 2023), and TBtools software was used to perform BLAST to compare Arabidopsis *WOX* gene family protein sequences with foxtail millet (threshold: 1 × 10^−5^), identifying *SiWOX* genes. *SiWOX* genes obtained from both methods were subjected to domain prediction using the Pfam website, selecting genes containing the HD domain as candidate genes, and finally determining the *SiWOX* gene family members. Published gene numbers were used to obtain WOX protein sequences of maize, rice, tomato, and Brachypodium distachyon. The MEGA (v11.0) [56] software was used to align WOX protein sequences of Arabidopsis (16), tomato (10), rice (13), maize (21), foxtail millet (13), and green foxtail (13) and construct a phylogenetic tree using the neighbor-joining method (bootstrap = 1000). The resulting tree was beautified using iTOL (https://itol.embl.de/, accessed on 26 December 2023).

### 2.2. Protein Properties and Sequence Analyses of SiWOX Genes

The physicochemical properties of SiWOX proteins, including the number of amino acids, molecular weight, isoelectric point, instability index, aliphatic index, and hydropathy index, were analyzed using the ExPASy website (https://web.expasy.org/protparam/, accessed on 5 December 2023). Subcellular localization prediction of SiWOX protein sequences was performed using the website (http://www.csbio.sjtu.edu.cn/bioinf/plant-multi/#, accessed on 5 December 2023). Protein spatial structure prediction of SiWOX was conducted using the protein structure database website (https://alphafold.ebi.ac.uk/, accessed on 5 December 2023).

### 2.3. Sequence Alignment, Gene Structure, and Conserved Motif Analysis of SiWOX Gene Family Members

SiWOX protein sequences were aligned using the CLUSTALW website (https://www.ebi.ac.uk/Tools/msa/clustalo/, accessed on 18 December 2023). The alignment results were downloaded and visualized using ESPript (https://espript.ibcp.fr/ESPript/cgi-bin/ESPript.cgi, accessed on 18 December 2023). The exon and intron structures of SiWOX were analyzed using GSDS2.0 (http://gsds.gao-lab.org/, accessed on 19 December 2023). Conserved motifs of *SiWOX* genes were predicted using the MEME online website (https://meme-suite.org/meme/, accessed on 25 December 2023), with the number of motifs set to 12, the minimum amino acid residues set to 3, the maximum amino acid residues set to 70, and other parameters set to default.

### 2.4. Analysis of Promoter Cis-Acting Elements of SiWOX Gene Family Members

The upstream 2000 bp sequences of *SiWOX* genes were extracted using TBtools software and submitted to the PlantCARE online website (http://bioinformatics.psb.ugent.be/webtools/plantcare/html/, accessed on 5 December 2023) for prediction of promoter *cis*-acting elements.

### 2.5. Chromosomal Localization and Collinearity Analysis of SiWOX Gene Family Members

Chromosomal position data of *SiWOX* were obtained from the gene annotation file. Using TBtools’ One Step MCScanX with default parameters, genes undergoing tandem duplication were identified, and Advanced Circos was used for visualization. Similarly, One Step MCScanX was used to identify collinear genes between foxtail millet and green foxtail, rice, and maize. The results were integrated using File Merge For MCScanX, and Multiple Synteny Plot was used to draw inter-species collinearity diagrams.

### 2.6. Analysis of Expression Patterns of SiWOX Gene Family Members

Transcriptome data of different developmental stages and tissues of Yugu1 were obtained from the *Setaria-db* database (www.setariadb.com/millet, accessed on 8 January 2023) [57], The tissue naming method followed the study by Meng et al. Heatmaps were generated using TBtools to compare differences in *SiWOX* expression. In the experimental field of Shanxi Agricultural University’s Minor Crops molecular Breeding Team, Jingu21 (provided by the SXAU Minor Crops Functional Genomics Center) was planted. At the 11–12 leaf stage of foxtail millet, the first period of cone panicles (approximately 1.0–1.5 mm) was sampled, and at the 13–14 leaf stage, the second period of panicles (approximately 2.5–3.0 mm) was sampled. Each sample had 3 biological replicates. The Yugu1 (provided by SXAU Minor Crops Functional Genomics Center) were hydroponically cultivated in an artificial climate chamber at Shanxi Agricultural University (light intensity: 50,000 LX; 16 h of daylight at 28 °C, 8 h of night at 22 °C) with ABA (2 μM) and distilled water as controls. After 9 days, the roots of seedlings were sampled, with 3 biological replicates per sample. After freezing with liquid nitrogen, the panicles and roots were sent to Novogene Bioinformatics Technology Co., Ltd. (Beijing, China) for high-throughput sequencing on the Illumina Hiseq platform. The sequencing results were analyzed using the R language to generate bar charts.

### 2.7. Response of SiWOX to Plant Hormones

Jingu21 seedlings were hydroponically cultivated in a growth chamber at the College of Life Sciences, Shanxi Agricultural University, using Hoagland’s solution (Beijing Coolaber Technology Co., Ltd., Beijing, China). The light intensity was set to 50,000 LX, with a photoperiod of 16 h of light at 28 °C during the day and 8 h of darkness at 22 °C during the night. After 28 days of foxtail millet germination, seedlings were treated with six hormones: 6-benzylaminopurine (6-BA) (100 µM), abscisic acid (ABA) (100 µM), salicylic acid (SA) (100 µM), gibberellic acid (GA3) (100 µM), jasmonic acid (JA) (100 µM), and indole-3-acetic acid (IAA) (100 µM). Leaf samples were taken at 0 h, 0.5 h, 2 h, 6 h, and 12 h after treatment. Each sample had three biological replicates and was frozen in liquid nitrogen and stored at −80 °C. Primers were designed using the foxtail millet multi-omics database MDSi (http://foxtail-millet.biocloud.net/home, accessed on 4 December 2023) and synthesized by Sangon Biotech Co., Ltd. (Shanghai, China) (Appendix A) with specificity checked. The gene *Si9g37480* was used as an internal reference [43]. Total RNA was extracted from the leaves using RNA extraction reagent (Beijing Coolaber Technology Co., Ltd., Beijing, China) following the Trizol method. Real-time quantitative polymerase chain reaction (qPCR) was performed using the UnionScript First-strand cDNA Synthesis Mix for qPCR (with dsDNase) for reverse transcription. Before amplification, the cDNA template was diluted fivefold. The qPCR reaction system was prepared using ChamQ^TM^ Universal SYBR qPCR Master Mix (Vazyme Biotech Co., Ltd., Nanjing, China). All qPCR reactions were performed on a Bio-Rad CFX96 Touch qPCR instrument, with each reaction repeated three times. The relative expression levels were calculated using the 2^−ΔΔCT^ method [58], and line graphs were generated using R language.

## 3. Results

### 3.1. Identification and Phylogenetic Analysis of SiWOX Gene Family

To identify *WOX* genes in foxtail millet, we conducted BLAST and HMM searches using the 13 *WOX* genes from Arabidopsis. A total of 13 putative *WOX* genes were identified in foxtail millet, the same number as in rice and green foxtail [47]. Structural domain prediction using Pfam revealed that all 13 foxtail millet *WOX* genes contain the HD domain. Based on their chromosomal distribution order, they were named *SiWOX1* to *SiWOX13*. To analyze the evolutionary pattern of the *WOX* gene family in foxtail millet, we performed cluster analysis and constructed a phylogenetic tree using protein sequences of WOX genes from Arabidopsis (16), tomato (10), rice (13), maize (21), foxtail millet (13), and green foxtail (13) (Figure 1). The 86 WOX proteins from these six species were divided into three evolutionary clades, consistent with previous classification results [3,7,47]. Among them, the modern clade had the most members (46), followed by the intermediate clade (29), and the ancient clade (11). In foxtail millet, *SiWOX2*, *SiWOX3*, *SiWOX7*, *SiWOX8*, *SiWOX10*, *SiWOX11,* and *SiWOX12* belonged to the modern clade; *SiWOX1*, *SiWOX4*, *SiWOX5*, *SiWOX9*, and *SiWOX13* belonged to the intermediate clade; only *SiWOX6* belonged to the ancient clade. This distribution indicates conserved functional characteristics in foxtail millet growth and development. Further analysis revealed that *SiWOX2*, along with *WUS* from maize, Arabidopsis, and tomato, formed a separate cluster, indicating the conservation of WUS protein function in these species. *SiWOX2* is speculated to play a role in SAM differentiation during foxtail millet growth and development. Additionally, because foxtail millet, green foxtail, maize, and rice are monocotyledonous plants, while Arabidopsis and tomato are dicotyledonous plants, the genes of foxtail millet always clustered with those of green foxtail, maize, and rice, indicating a closer phylogenetic relationship with them than with Arabidopsis and tomato. Notably, the number of *WOX* genes in foxtail millet is the same as that in green foxtail, and their relationship in the evolutionary tree is the closest, suggesting that each pair of genes in foxtail millet and green foxtail may be orthologous genes, supporting the hypothesis that foxtail millet was domesticated from green foxtail [59].

### 3.2. Structure Characterization of the WOX Proteins of Foxtail Millet

Analysis of the characteristics of foxtail millet WOX proteins (Table 1) reveals that the 13 foxtail millet WOX proteins consist of 212 to 531 amino acids, with relative molecular weights ranging from 24.01 to 54.80 kDa. The theoretical isoelectric points range from 6 to 9.26, with most proteins being alkaline, while only 3 are acidic (SiWOX4, SiWOX6, and SiWOX13). The instability coefficients range from 50.26 to 78.12, with SiWOX6 being the only member belonging to the ancient clade and its protein being the most stable. The protein hydropathy coefficients are all negative, indicating that these family members are hydrophilic proteins.Since WOX proteins are transcription factors, subcellular localization results predict that all SiWOX proteins are located in the cell nucleus. Analysis of the predicted protein spatial structures of SiWOX proteins reveals that all SiWOX proteins have similar spatial structures (Figure 2). Among them, SiWOX1 and SiWOX13, SiWOX4 and SiWOX5, and SiWOX11 and SiWOX12 have the most similar protein structures, and they are also clustered into the same subfamily in the evolutionary tree. Detailed information about the amino acids of SiWOX proteins is provided in Appendix A.

### 3.3. Sequence Alignment, Gene Structure, and Conserved Motif Analysis of SiWOX Gene Family Members

Visualization of the multiple sequence alignment results of identified SiWOX protein amino acids (Appendix A) reveals that all SiWOX proteins contain a conserved homologous domain composed of 60 amino acid residues, known as the HD. This number of amino acid residues in the homologous domain is consistent with that found in WOX proteins from rice, Arabidopsis, maize, and sorghum [47]. SiWOX2 has an additional tyrosine (Y) residue in its homologous domain compared to other SiWOX proteins. Previous studies have shown that the additional tyrosine residues in the homologous domains of WUS proteins from Arabidopsis, maize, rice, and sorghum are highly conserved [47], further indicating that *SiWOX2* likely functions similarly to the *WUS* gene in foxtail millet, Arabidopsis, maize, rice, and sorghum, primarily maintaining the number and characteristics of stem cell populations during foxtail millet SAM differentiation [12]. This also suggests that this residue may play an important role in the functionality of WUS transcription factors. Within the homologous domains of SiWOX, there are 17 highly conserved amino acid residues, consistent with previous studies [47]. All modern clade members of SiWOX proteins contain a WUS-box domain, with the amino acid sequence present in the form of T-L-X-L-F-P-X-X (where X represents any amino acid). Previous studies have shown that this domain is unique to modern clade members [3]. In Arabidopsis WOX transcription factors, WUS, WOX5, and WOX7 contain a conserved ethylene-responsive element binding factor-associated amphiphilic repression (EAR)-like motif at their C-terminus, the main function of which is to suppress downstream gene expression [3], *SiWOX2* and *SiWOX8* also contain an EAR-like motif at their carboxyl terminus, with the amino acid sequence L-E-L-X-L-X (where X represents any amino acid). In the phylogenetic tree, *SiWOX2* clusters with *AtWUS*, while *SiWOX8* clusters with *AtWOX5* and *AtWOX7*, suggesting that *SiWOX2* and *SiWOX8* in foxtail millet may have similar functions to *AtWUS*, *AtWOX5,* and *AtWOX7*, which are homologous to Arabidopsis.

In the phylogenetic evolution tree of foxtail millet WOX, *SiWOX* genes with similar gene structures and identical conserved motifs often cluster into one subfamily (Figure 3A). Gene structure analysis results (Figure 3B) show that *SiWOX2* and *SiWOX10* each contain only one intron, while the remaining *SiWOX* genes contain one intron in both the N-terminal and C-terminal regions. *SiWOX3*, *SiWOX7*, *SiWOX8*, *SiWOX11*, and *SiWOX12* contain two exons, while *SiWOX1*, *SiWOX2*, *SiWOX5*, *SiWOX6*, *SiWOX9*, *SiWOX10*, and *SiWOX13* contain three exons, and only *SiWOX4* contains four exons. Conserved motif prediction results (Figure 3C) identified 12 conserved motifs among the 13 *SiWOX* genes, with *SiWOX1* and *SiWOX13* containing the most conserved motifs, with a total of 7. All SiWOX proteins contain motifs 1 and 2, indicating that motifs 1 and 2 are conserved in SiWOX proteins, with the ancient clade *SiWOX6* being the most conserved protein, containing only motifs 1 and 2. Motif 1 and motif 2 correspond to the HD domain of SiWOX. Motif 4 includes motif 11, with motif 11 corresponding to the WUS-box domain. Motif 11 is only present in *SiWOX2*, *SiWOX3*, *SiWOX7*, *SiWOX8*, and *SiWOX10*, while motif 4 is only present in *SiWOX11* and *SiWOX12*. The WUS-box is only present in the modern clade, and these proteins belong to the modern clade in the evolutionary tree. Intermediate clade SiWOX proteins all contain motif 3, with *SiWOX4* and *SiWOX5* also containing motif 10, and *SiWOX1*, *SiWOX9*, and *SiWOX13* all containing motifs 5, 7, and 9. In addition, *SiWOX11* and *SiWOX12* uniquely contain motif 8.

### 3.4. Promoter Cis-Acting Element Analysis of SiWOX Gene Family Members

To further elucidate the potential functions of the *SiWOX* gene family in foxtail millet growth, development, and response to abiotic stresses, upstream 2000 bp sequences of *WOX* genes in foxtail millet were extracted for analysis using the PlantCare database (Appendix A). The results indicated an uneven distribution of various functional *cis*-acting elements on each *SiWOX* gene, primarily including two major categories: Stress response and plant hormone response *cis*-acting regulatory elements. *SiWOX9* contained the highest number of *cis*-acting elements, with 114, while *SiWOX2* only contained 30 *cis*-acting elements. All *SiWOX* promoters contained light-responsive elements, except for *SiWOX13*, suggesting that the remaining *SiWOX* genes may be associated with environmental stresses, as they contained *cis*-acting elements related to low temperature, drought, defense, and stress. *Cis*-acting elements related to hormone responses in *SiWOX* included methyl jasmonate (50), abscisic acid (36), gibberellin (13), auxin (6), and salicylic acid (3). The ancient clade *SiWOX6* had all types of *cis*-acting elements, while *SiWOX2* had only one salicylic acid-responsive *cis*-acting element, with the other two distributed in *SiWOX6* and *SiWOX7*. *SiWOX1*, *SiWOX6*, *SiWOX7*, *SiWOX9*, *SiWOX10*, and *SiWOX11* each had one auxin-responsive *cis*-acting element, suggesting that these genes are involved in auxin regulatory feedback during foxtail millet growth and development. Additionally, *SiWOX1*, *SiWOX2*, *SiWOX4*, *SiWOX5*, *SiWOX9*, *SiWOX10*, *SiWOX12*, and *SiWOX13* all contained *cis*-acting elements involved in seed-specific regulation, with *SiWOX2* and *SiWOX4* also containing *cis*-acting elements involved in endosperm-specific expression. These results suggest that *SiWOX* genes may be associated with foxtail millet stress resistance and hormone response pathways.

### 3.5. Chromosomal Localization and Collinearity Analysis of SiWOX Gene Family Members

To further analyze the evolutionary relationships of foxtail millet WOX family members, the identified foxtail millet *WOX* genes were mapped onto chromosomes based on gene annotation information, and the results were visualized using Advanced Circos (Figure 4). Except for chromosome1 and chromosome4, *WOX* genes were distributed on all other chromosomes, with chromosome5 having the most with four *WOX* genes, and chromosome2, chromosome6, and chromosome9 each having one *WOX* gene. Gene tandem duplication is an important internal driving force for crop domestication and breeding, playing a significant role in plant phenotypic variation [60,61,62]. Using MCScanX, the tandem repeat relationship of *WOX* genes in foxtail millet was analyzed, revealing that there are three pairs of tandem repeats in foxtail millet *WOX* genes, namely *SiWOX1* and *SiWOX13*, *SiWOX4* and *SiWOX5*, and *SiWOX11* and *SiWOX12*. Tandem repeats of *SiWOX11* and *SiWOX12* occurred in the modern clade, while the other two pairs occurred in the middle clade, with the ancient clade being the most conservative and having no gene tandem repeats. Gene tandem repeats can lead to changes in gene structure [63], such as *SiWOX4* having one more intron than *SiWOX5*, indicating that gene tandem duplication enriches the diversity of the *SiWOX* gene family.

Using the multiple synteny plot, a collinearity graph between green foxtail, foxtail millet, maize, and rice was generated (Figure 5). From the graph, it can be observed that the collinearity of *WOX* genes between green foxtail and foxtail millet is the highest, further suggesting that they may share a common ancestor. The collinearity of *WOX* genes between foxtail millet and maize is higher than that between foxtail millet and rice, possibly because foxtail millet and maize both belong to C_4_ crops, while rice is a C_3_ crop.

### 3.6. Analysis of the Expression Patterns of SiWOX Genes

Transcriptome data from various tissues at different growth and development stages of Yugu1 were obtained from the *Setaria-db* database, and a heatmap of gene expression was generated (Figure 6). The ancient clade member *SiWOX6* exhibited the most conservative function, with expression observed across different tissues at various stages of foxtail millet development. Among the intermediate clade members, *SiWOX1* was expressed in all other tissues at various stages except for anthers, mainly during the vegetative growth and seed maturation stages of foxtail millet, suggesting that its evolutionary rate is slower compared to other intermediate clade genes. Genes clustered in the same clade of the phylogenetic tree often share similar functions, but differences exist. For instance, *SiWOX11* and *SiWOX12* were primarily expressed in the later stages of stem apex meristem development, with *SiWOX12* also showing expression during the spikelet primordia and flowering stages. Additionally, expression of *SiWOX4* decreased initially and then increased with seed maturation, while *SiWOX5* exhibited the opposite trend, indicating distinct functions during foxtail millet seed maturation. *SiWOX1*, *SiWOX6*, *SiWOX8*, and *SiWOX13* were highly expressed in germinating seeds, suggesting their involvement in seed germination. *SiWOX1* and *SiWOX13* showed high expression levels in various parts of foxtail millet such as roots, stems, and leaves, indicating their association with cell proliferation, division, differentiation, and tissue elongation. *SiWOX10* exhibited high expression levels in the stems and nodes of foxtail millet and contained *cis*-acting elements responsive to auxin, suggesting its involvement in cytokinin and auxin synthesis. *SiWOX1* and *SiWOX6* were highly expressed in primary branching during flowering, indicating their joint influence on inflorescence development in foxtail millet. As foxtail millet seeds matured, expression of *SiWOX9* gradually decreased, indicating its involvement in early seed development. Only *SiWOX2* exhibited tissue specificity, with high expression only during the development of stem apical meristems, suggesting its role in controlling panicle development in foxtail millet. These results suggest that *SiWOXs* may play important roles in the vegetative and reproductive growth processes of foxtail millet.

The development of the foxtail millet panicle involves a continuous differentiation process, starting from the SAM forming a “smooth round growth cone”, which elongates and develops small protrusions to form primary branches. As primary branches grow, they initiate the differentiation of secondary branches, and after tertiary branch differentiation, spikelets and florets begin to differentiate at the tip of tertiary branches, completing panicle development [64]. Analysis of transcriptome sequencing results of foxtail millet panicle development (Figure 7A) revealed higher expression levels of *SiWOX2*, *SiWOX3*, *SiWOX4*, *SiWOX5*, and *SiWOX6* compared to *SiWOX1*, *SiWOX10*, *SiWOX11*, and *SiWOX12*. Among them, the ancient clade member *SiWOX6* exhibited the highest expression level, followed by *SiWOX3* and *SiWOX5*, while *SiWOX7*, *SiWOX8*, *SiWOX9*, and *SiWOX13* showed almost negligible expression levels. During the second stage of panicle development, the expression levels of *SiWOX3* and *SiWOX10* significantly increased, while *SiWOX5* decreased, and the expression levels of other genes remained almost unchanged. In foxtail millet seedling roots, fewer *SiWOX* genes were expressed compared to panicle development, but their expression levels were relatively high (Figure 7B). Among them, *SiWOX6* exhibited the highest expression level, followed by *SiWOX13*, *SiWOX8*, *SiWOX1*, and *SiWOX9*. ABA plays a crucial role in regulating plant root growth and development [65]. Compared to the control, the expression levels of *SiWOX6* and *SiWOX13* were significantly upregulated in seedling roots treated with ABA, while *SiWOX1* and *SiWOX9* also showed upregulation, and *SiWOX8* was downregulated. Overall, many members of the *SiWOX* gene family are specifically involved in the development of panicle and root in foxtail millet.

### 3.7. SiWOX Gene Response to Plant Hormones

To investigate the response of *SiWOX* genes to various hormones, 28-day-old foxtail millet seedlings were treated with six plant hormones: 6-BA (100 μM), ABA (100 μM), GA3 (100 μM), SA (100 μM), MeJA (100 μM), and IAA (100 μM). The expression patterns of *SiWOX1*, *SiWOX2*, *SiWOX6*, *SiWOX7*, *SiWOX10*, and *SiWOX13* genes, distributed across different subfamilies in foxtail millet seedling leaves, were analyzed (Figure 8). Results showed that after treatment with MeJA, all six *SiWOX* genes exhibited an up-down-up-down expression pattern, with expression peaking at 0.5 h after treatment. By 12 h after treatment, except for *SiWOX7*, the expression levels of the other five *SiWOX* genes decreased to levels observed at 2 h after treatment. Except for *SiWOX7* and *SiWOX10*, the expression levels of the remaining four *SiWOX* genes remained nearly unchanged in response to GA treatment. Compared to the other four hormones, the ancient clade member *SiWOX6* showed minimal response to ABA and GA. *SiWOX1* demonstrates analogous trends in response to JA and SA. The relative expression level at 0.5 h post-hormone treatment surpasses that at 12 h by more than double. Conversely, under IAA treatment, the relative expression level at 0.5 h is approximately half of that observed at 12 h. Additionally, *SiWOX10* and *SiWOX13* exhibited similar trends in response to the six hormones. These results suggest that the *SiWOX* gene family may have different potential functions during foxtail millet leaf development.

## 4. Discussion

Foxtail millet holds significance as an important cereal crop due to its resilience to drought, tolerance to poor soil, and salinity, characterized by well-developed roots [66,67,68]. WOX transcription factors, unique to plants, play a pivotal and conserved role in plant growth, development, response to various abiotic stresses, and plant hormone signaling [69,70,71]. While extensively studied in plants such as Arabidopsis, maize, rice, and tomato [8,47,72], the *WOX* gene family in foxtail millet has not been previously reported. This study employed two methods and conservative domain prediction to identify 13 *WOX* genes in foxtail millet (Table 1), consistent with the number in green foxtail and rice, as evidenced by phylogenetic analysis (Figure 1), underscoring the highly conserved distribution of unique clades (ancient, intermediate, and modern) within this family in foxtail millet. Previous studies on WOX family members have primarily focused on evolution. The foxtail millet *WOX* gene family exhibits only one member in the ancient clade, with the maximum number of members in the modern clade being seven, akin to findings in other species [47]. Unlike the AtWUS protein sequence, which features a non-conservative acidic amino acid sequence between the HD homologous domain and the WUS-box domain [47], the multiple sequence alignment results (Appendix A) indicate the absence of such an amino acid sequence in the SiWOX protein sequence, suggesting that the acidic region domain may predominantly influence Arabidopsis WOX transcription factors.

In the phylogenetic tree (Figure 1), *SiWOX2* clusters with *WUS* in maize, Arabidopsis, and tomato, forming a distinct cluster. Through multiple sequence alignment of the foxtail millet *WOX* gene family (Appendix A), we identified additional tyrosine (Y) residues in the homologous domain of *SiWOX2*, which are highly conserved with the additional tyrosine (Y) residues in the *WUS* domain of Arabidopsis, rice, maize, and tomato [47]. The main function of *AtWUS* is to maintain the number and characteristics of SAM stem cells [12]. In the gene expression pattern (Figure 6), *SiWOX2* shows specific expression in the SAM, suggesting a similar function in regulating the size of the SAM in foxtail millet. In maize *BIF3* mutants, overexpression of *ZmWUS1* leads to spherical spikes and a significant reduction in yield [73]. Controlling the expression of meristem genes can alter plant yield without affecting the growth of explants [74], highlighting the potential importance of *SiWOX2* in improving foxtail millet yield traits. Furthermore, collinearity analysis (Figure 5) reveals a high degree of collinearity between the WOX family members of foxtail millet and green foxtail. The differences between the inflorescences of foxtail millet and green foxtail, such as changes in the number, density, and order of primary branches [75], may be related to differences in the *WUS* gene between the two species, warranting further investigation.

In Arabidopsis, rice, maize, and tomato, different members of the *WOX* gene family have distinct functions, and phylogenetic tree studies provide references for predicting the functions of *WOX* genes in foxtail millet. The *WOX* gene family plays a crucial role in regulating plant growth and development processes [76,77,78]. The *CLV*-*WUS* feedback pathway ensures the self-renewal of stem cells while forming new plant tissues and organs [79]. Recently, the foxtail millet *CLE* gene family has also been identified and reported [45]. In the evolutionary tree (Figure 1), *SiWOX2*, together with *AtWUS*, *ZmWUS1*, *ZmWUS2*, *SvWUS*, and *SlWUS*, forms a subcluster and is specifically expressed during foxtail millet inflorescence meristem development, indicating the possible existence of a *CLV*-*WUS* feedback pathway in the SAM of foxtail millet, with *SiWOX2* playing a crucial role in the signal transduction of the SAM *CLV*-*WUS* feedback. *AtWOX5* is specifically expressed in the root apical meristem, and similar regulatory pathways to those in the SAM also exist in Arabidopsis root apical meristem [23,24]. In the evolutionary tree, *SiWOX8* clusters with *AtWOX5* in the same subcluster; in the gene expression pattern (Figure 6), *SiWOX8* shows higher expression levels in the roots of foxtail millet seedlings and at the one-leaf-two-heart stage, indicating that *SiWOX8* may play an important role in the development of the root apical meristem in foxtail millet. *OsWOX11* is a key regulator of crown root development in rice, and its expression is induced by auxin and cytokinin. Recently, a dynamic feedback regulation mechanism formed by *WOX11*-*JMJ706*-*LBD16* was discovered in the growth process of rice crown roots, ensuring that *LBD16* maintains an appropriate expression level to ensure the growth of rice crown roots [80]. In the evolutionary tree (Figure 1), *SiWOX1* and *OsWOX11* cluster together, and combined with the analysis of *SiWOX* gene expression patterns in different tissues (Figure 6) and the expression pattern of *SiWOX* genes in the roots of foxtail millet (Figure 7B), we found that *SiWOX1* is highly expressed in the roots of foxtail millet at different stages, indicating its important role in the root development of foxtail millet.

## 5. Conclusions

This study provides a comprehensive analysis of the foxtail millet *SiWOX* gene family, with 13 *SiWOX* genes divided into three subclusters. All SiWOX proteins contain a conserved HD domain, with SiWOX2 featuring an additional tyrosine (Y) residue in this domain. *SiWOX* genes clustered in the same clade of the phylogenetic tree exhibit similar protein spatial structures, gene structures, and conserved motifs. Tandem repeats analysis reveals three pairs of repeats among the *SiWOX* genes, while collinearity analysis shows the highest collinearity between foxtail millet and green foxtail, suggesting a common ancestor. Analysis of *SiWOX* expression patterns and hormone responses reveals potential functional diversity among different *SiWOX* genes in foxtail millet. Overall, this study provides a foundation for further elucidating the functions of *SiWOX* genes in foxtail millet.

## Figures and Tables

**Figure 1 genes-15-00476-f001:**
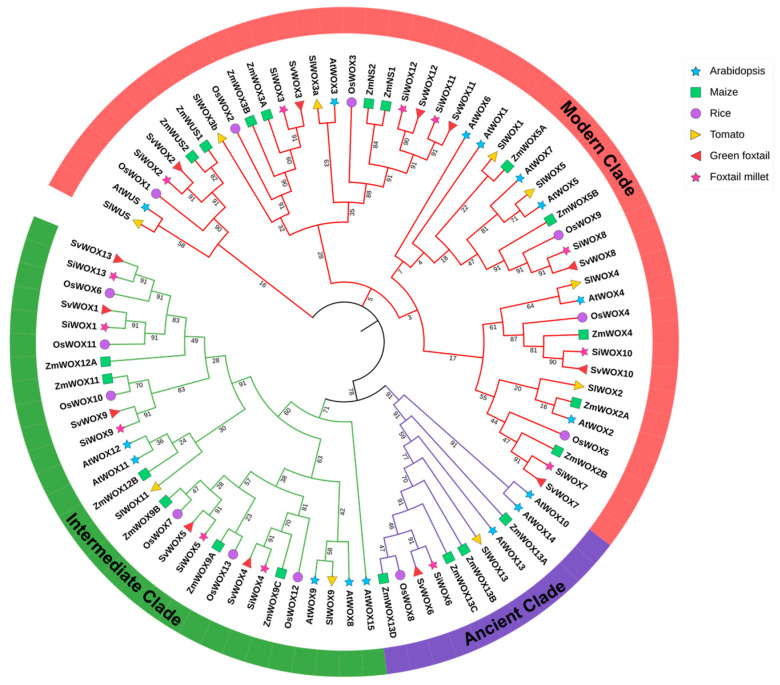
Phylogenetic analysis of WOX proteins of foxtail millet, green foxtail, tomato, maize, rice, and Arabidopsis. A phylogenetic evolutionary tree was constructed using MEGA-11, where clades of the same color represent that they belong to the same subfamily. The *WOX* gene IDs of the six species are supplemented in Appendix A. Node numbers: bootstrap values, a put back sampling statistical method used to test the credibility of evolutionary tree branches.

**Figure 2 genes-15-00476-f002:**
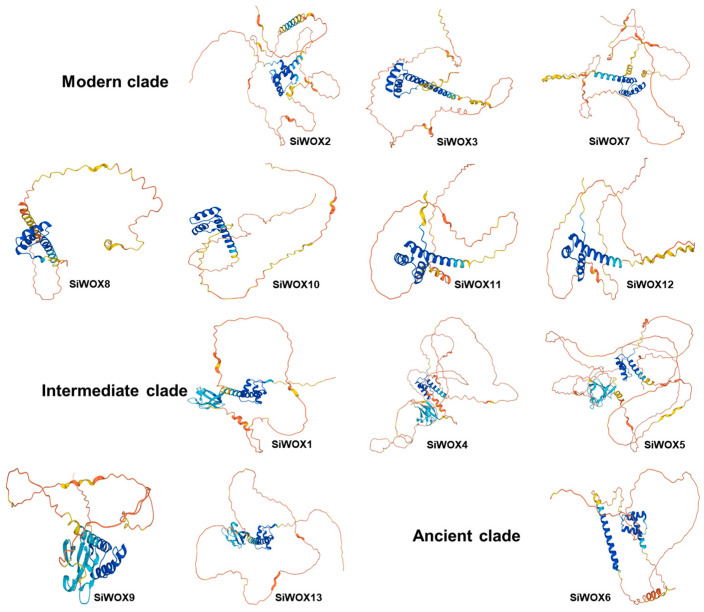
Prediction of the spatial structure of SiWOX proteins. Modern clade: SiWOX2, SiWOX3, SiWOX7, SiWOX8, SiWOX10, SiWOX11, and SiWOX12; Intermediate clade: SiWOX1, SiWOX4, SiWOX5, SiWOX9, and SiWOX13; Ancient clade: SiWOX6. The spatial structure predictions of SiWOX proteins links: https://alphafold.ebi.ac.uk/ (accessed on 5 December 2023).

**Figure 3 genes-15-00476-f003:**
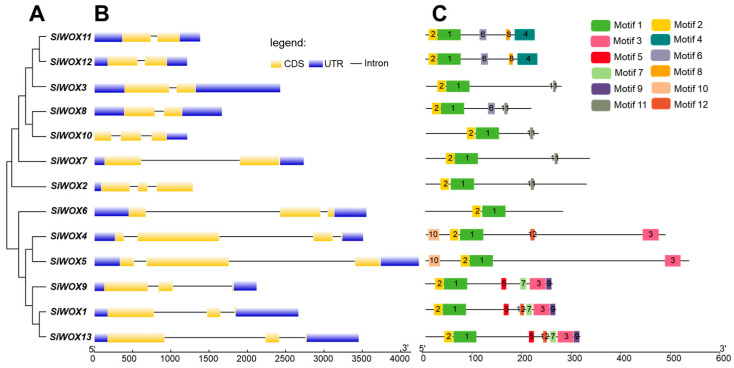
Analysis of *SiWOX* gene structure and conserved motifs. (**A**) Phylogenetic tree. (**B**) Gene structure. Coding sequences (CDS) and untranslated regions (UTRs) are represented by different colored boxes, while introns are indicated by lines. (**C**) Conserved motifs. Conserved motifs within the *SiWOX* genes are represented by boxes of different colors. The weblogo diagram of the twelve conserved motifs is provided in Appendix A.

**Figure 4 genes-15-00476-f004:**
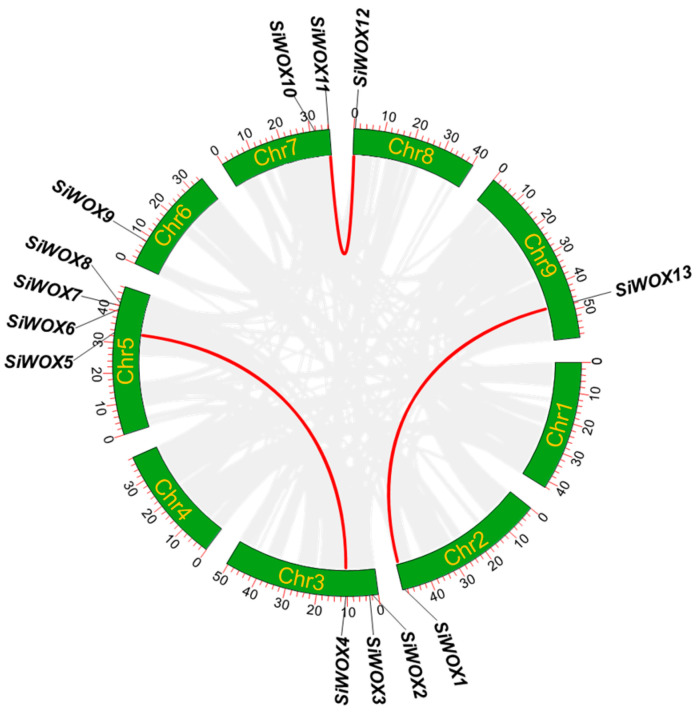
Chromosome distribution and gene tandem repeats of *SiWOX* genes. Gray lines represent tandem repeats between foxtail millet genomes, while red lines represent tandem repeats between *SiWOX* genes.

**Figure 5 genes-15-00476-f005:**
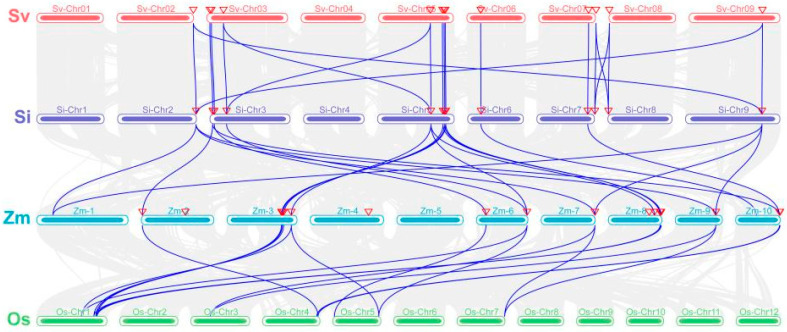
Collinearity analysis of *WOX* genes in *Setaria viridis*, *Setaria italica*, *Zea mays*, and *Oryza sativa*. Gray lines indicate collinearity between genes from different species, while purple lines indicate collinearity between WOX genes from different species. Sv: *Setaria viridis*, Si: *Setaria italica*, Zm: *Zea mays*, Os: *Oryza sativa*.

**Figure 6 genes-15-00476-f006:**
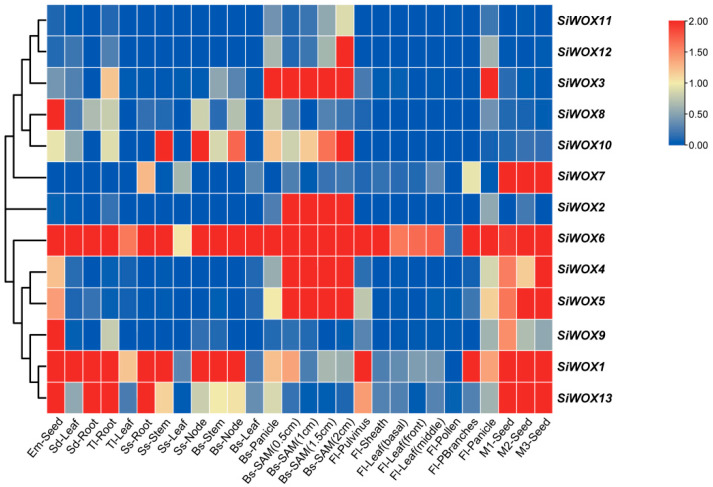
Expression pattern analysis of *SiWOX* genes in different tissues. The heatmap of the expression profiles of the *SiWOX* gene in different developmental stages is represented by normalized values using RNA-seq data, with the color from red to blue indicating the expression levels from high to low. Tissue information for *SiWOX* expression analysis is provided in Appendix A.

**Figure 7 genes-15-00476-f007:**
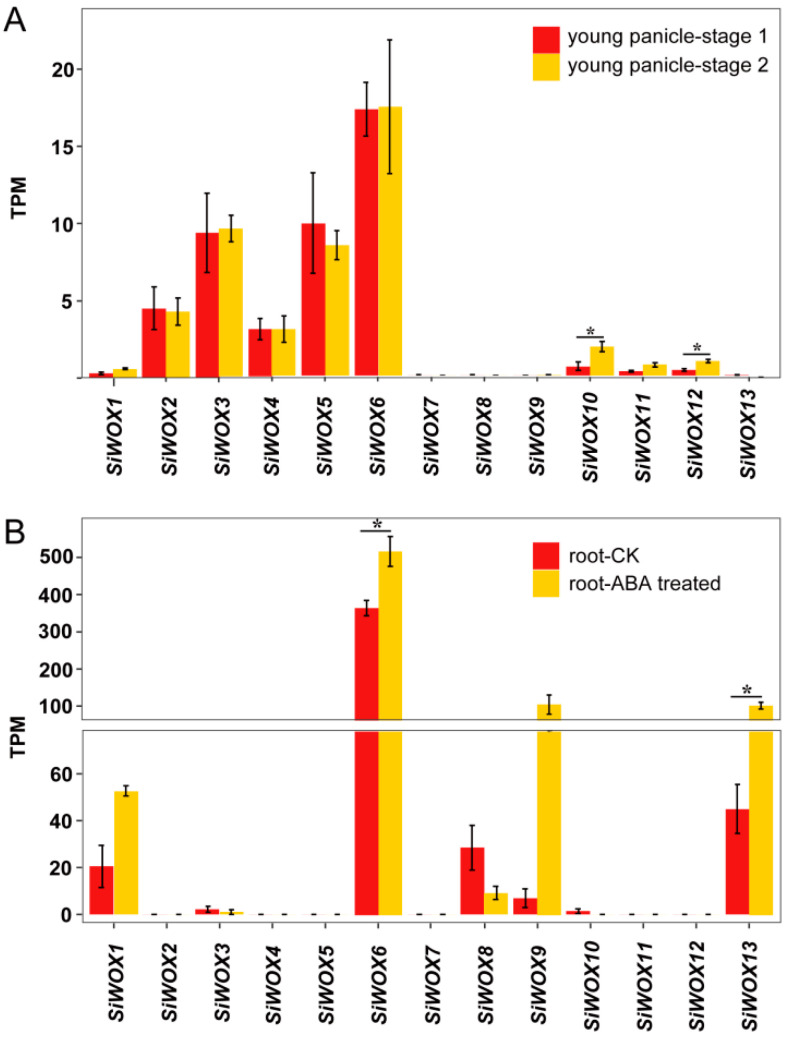
Expression patterns of *SiWOX* genes in foxtail millet panicles and roots. (**A**) Expression patterns of *SiWOX* genes in panicles at different developmental stages: Stage 1, panicle length approximately 1.0–1.5 mm; stage 2, panicle length approximately 2.5–3.0 mm. (**B**) Expression patterns of *SiWOX* genes in 9-day-old seedling roots treated with ABA (2 μM) and without ABA (CK). Bar graphs represent differences in transcriptome sequencing duplicates between spike samples (**A**) and root samples (**B**). Statistical significance between panicle (**A**) and root (**B**) stages was determined by *t*-test (* *p* < 0.05). TPM: transcripts per million.

**Figure 8 genes-15-00476-f008:**
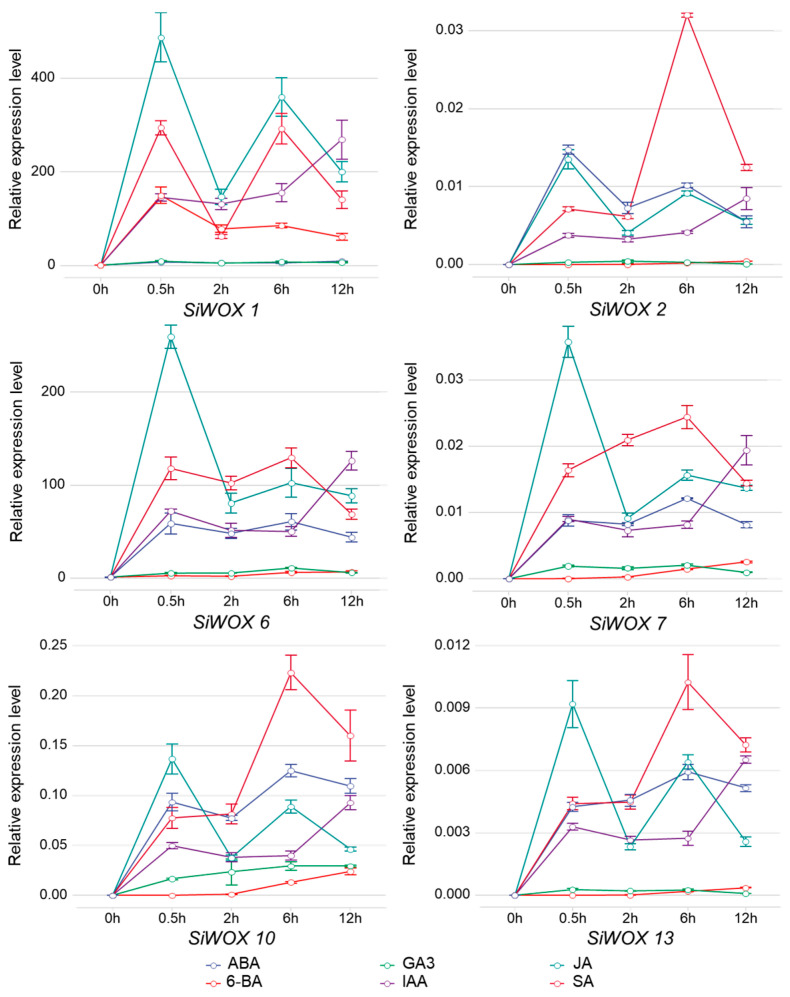
Expression pattern of *SiWOX* genes in response to plant hormones. Six plant hormones, 6-BA (100 μM), ABA (100 μM), GA3 (100 μM), SA (100 μM), MeJA (100 μM), and IAA (100 μM), were used to treat 28-day-old seedlings, and leaf samples were taken at 0 h, 0.5 h, 2 h, 6 h, and 12 h post-treatment. The gene expression levels at different time intervals were detected using the qPCR method. Line graphs represent changes in gene expression levels.

**Table 1 genes-15-00476-t001:** The properties of SiWOX proteins.

Gene Name	Gene ID	Number of Amino Acids	Molecular Weight	Isoelectric Point	Instability Index	Aliphatic Index	GRAVY	Subcellular Localization
*SiWOX1*	*Si2g43190*	263	27.50	7.84	62.06	63.57	−0.264	Nucleus
*SiWOX2*	*Si3g03660*	325	33.49	8.61	60.8	62.06	−0.306	Nucleus
*SiWOX3*	*Si3g05200*	273	28.59	7.66	66.92	69.3	−0.137	Nucleus
*SiWOX4*	*Si3g14560*	483	50.91	6.92	67.92	82.11	−0.19	Nucleus
*SiWOX5*	*Si5g27130*	531	54.80	7.16	57.26	70.96	−0.2	Nucleus
*SiWOX6*	*Si5g36340*	278	30.91	6	50.26	59.42	−0.733	Nucleus
*SiWOX7*	*Si5g37740*	331	35.19	8.93	78.12	62.51	−0.419	Nucleus
*SiWOX8*	*Si5g38630*	212	24.07	8.36	52.12	68.07	−0.703	Nucleus
*SiWOX9*	*Si6g09430*	257	27.89	7.22	64.8	65.41	−0.352	Nucleus
*SiWOX10*	*Si7g26980*	228	25.39	9.26	66.32	74.87	−0.534	Nucleus
*SiWOX11*	*Si7g33510*	220	24.01	8.83	68.04	52.95	−0.673	Nucleus
*SiWOX12*	*Si8g01120*	225	24.55	8.98	71.07	51.33	−0.732	Nucleus
*SiWOX13*	*Si9g41130*	312	32.00	6.74	69	63.27	−0.249	Nucleus

## Data Availability

All data generated or analyzed during this study are included in this published article.

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
