# Peer review of "Identification and Expression Analysis of the WOX Transcription Factor Family in Foxtail Millet (Setaria italica L.)"

_genes, 2024, doi:10.3390/genes15040476_

Round 1

Reviewer 1 Report

Comments and Suggestions for Authors

The manuscript is very well written and structured. It is really enjoyable reading but some data and results should be improved, especially the gene expression analysis (wet lab experiment).

The introduction could be shorten, just emphisizing on the important role of WOX transcription factor family in plant growth and development. Some description can be moved to the Discussion section.

Lines 162-164: the reference for MEGA program should be included in the text:

Tamura, K., Stecher, G., & Kumar, S. (2021). MEGA11: molecular evolutionary genetics analysis version 11. Molecular biology and evolution, 38(7), 3022-3027.

Line 286: It is probably Figure 2 instead of Figure 3.

Figure 5: It would be better if it is more sharpen and with bigger font size

Figure 7: "TPM" should be explained

Line 470: Section 3.7. SiWOX Gene Response to Plant Hormones. Some figures about the difference of gene expression between samples taken at various time points should be provided. How many times expression in one samples is higher / lower than in others. It would improve the presentation of the results and help understand them better.

Figure 8: What are the units used as Relative expression level? Usually it is 2−ΔΔCT (times of difference between two samples; one sample should be taken as 1, others are compared to it).

Reviewer 2 Report

Comments and Suggestions for Authors

Dear Authors,

Reviewer comments genes-2930661

The manuscript entitled „identification and expression analysis of the WOX transcription factor family in foxtail millet (Setaria italica L.)“ represents the first original study aimed at gene and protein structure description, phylogenetic analysis, collinearity analysis, and expression analysis of SiWOX genes in foxtail millet. The manuscript thus provides a complex study on SiWOX genes.

However, I have a few comments on the present manuscript which are given below:

1/ In Materials and methods, all plant materials used for the experiments have to be listed including their sources, i.e., from which institutions were obtained. I tis evident that two genotypes of foxtail millet, Yugu 1 and Jingu21, were used for the experiments. The sources of both genotypes have to be given and it also has to be specified for which analyses the given genotypes were used.

2/ In Results, phylogenetic analysis, Figure 1, the numbers at nodes have to be explained in Figure 1 legend. Moreover, the titles of the three clades, i.e., ancient, intermediate, and modern, have to be briefly explained. Does the name „ancient“ mean the oldest one shared with green foxtail as a common ancestor? What are the major differences between these three clusters?

3/ In Results, Figure 2, correct the typing error in „Modern clade“ heading. Moreover, a short terminology vocabulary of the motifs shown should be added. A software used for spatial structure modeling should be named in Figure 2 legend.

4/ In Discussion, i tis stated that SiWOX2 in foxtail millet has an analogous function to AtWUS in Arabidopsis thaliana, i.e., to maintain the number and characteristics of shoot apical meristem cells. Does it mean that foxtail millet does not have WUSCHEL genes? The main differences between WUS and WOX genes have to be clearly given.

5/ Formal comments on the text:

Introduction, line 43: Add a space between the two sentences and two words „residue“ and „These“.

Introduction, line 131: Add a space between the reference „[46]“ and „families“.

Line 262: Add a space between the two sentences and two words „development“ and „Additionally“

Line 314: Add a space between the word „members“ and the following reference.

Line 343: The statement „….that SiWOX4 and SiWOX5, as well as SiWOX9, SiWOX1, and SiWOX13“ has to be completed since it lacks any verb and is not grammatically correct.

Line 361: Add a space between the two sentences and two words „stress“ and „Cis-acting elements“.

Figure 7 legend, line 466: Add a space between the number and teh corresponding unit in „2 μM“.

Line 477 and further text: Use small „h“ instead of capital „H“ for the time unit „hour“.

Comments on the Quality of English Language

-
